# Influence of Electromagnetic Interference on the evaluation of Lidar-derived Aerosol properties from Ny-Ålesund, Svalbard

Tim Poguntke[1,*] and Christoph Ritter[2,*]

[1]Kempten University of Applied Sciences, Bahnhofstraße 61, 87435 Kempten, Germany
[2]Alfred Wegener Institute, Telegrafenberg, Gebäude A45, 14473 Potsdam, Germany
[*]These authors contributed equally to this work.

**Correspondence:** Christoph Ritter (christoph.ritter@awi.de)

**Abstract.** Possible interference sources for our aerosol lidar setup with transient recorders have been assessed. This was done by two methods: a spectrum analysis of the lidar signals in order to detect radio-frequency interference and measurements of the electromagnetic interference caused by the laser power supply. We found disturbances in the analog channels of the transient recorders, presumably caused by ageing effects of our older recorders. An easy method on how the signal-to-noise-ratio can be improved retrospectively is presented. We also show that the usage of two-way radio at our location leads to a noticeable radio-frequency interference in the lidar profiles. Further, we present measurements of the electromagnetic interference caused by the laser power supply, which may lead to disturbances in the lidar profiles if the transient recorders are placed next to it.

## 1 Introduction

Lidar (i.e. light detection and ranging) is a mature technology for aerosol research since many years and it is already employed in dedicated networks like EARLINET (Pappalardo et al., 2014), AD-NET (Shimizu et al., 2016), LALINET (Guerrero-Rascado et al., 2016) and others. Hence, quality assurance and control will probably gain importance for long-term data recording. Freudenthaler et al. (2018) already discussed many aspects in this regard. If the understanding of disturbances and their sources in lidar signals gets improved, measurement equipment may be adapted as well as existing data sets may be improved retrospectively.

In this work, we analyze the noise increase in lidar signals provoked by electromagnetic interference (EMI) and how this worsens the derivation of aerosol properties. We present a spectral analysis of lidar signals in order to detect radio-frequency (RF) interference that decreases the signal-to-noise ratio (SNR). We also present an easy approach on how frequency-selective interference can be suppressed in order to increase signal quality retrospectively if the frequencies of the interference is known. We also provide measurements of the electromagnetic radiation of the power supply for the laser in order to address the following questions:

  i Are our lidar signals corrupted by RF interference?

  ii Are there other possible EMI sources?

  iii Does the laser power supply affect the recorders?

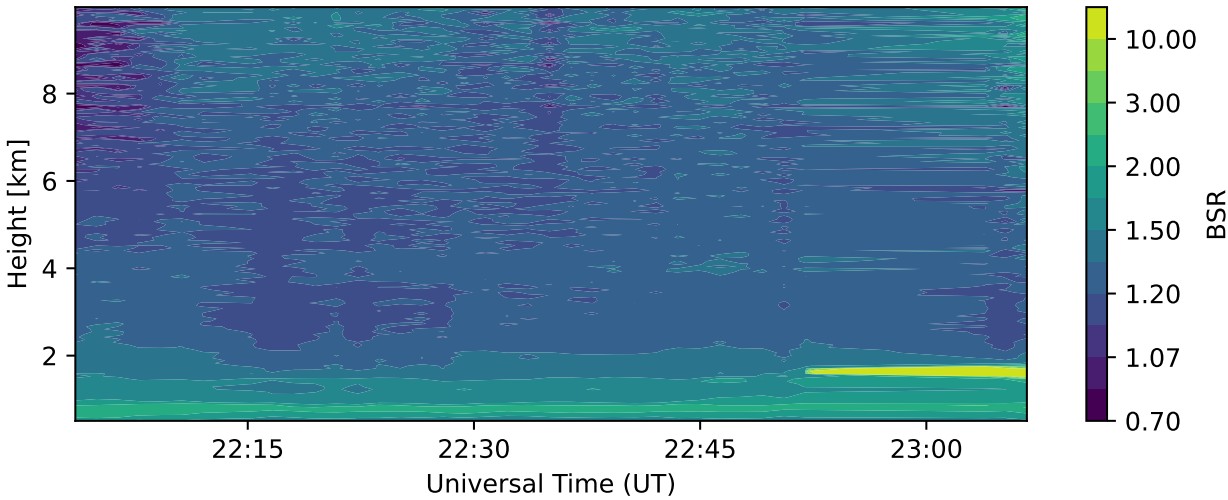

**Figure 1.** Overview of lidar observation on 16 Feb 2023. Presented is the backscatter ratio at 532 nm from the analog recording

However, de-noising techniques in lidar from a general point of view are outside the scope of this paper. Due to the strong
dependence of SNR on altitude such a noise filtering is commonly done by wavelet filtering (Zhou et al., 2013) or (Mao, 2012).
Instead, we will show that EMI can be suppressed in frequency domain, if it appears at fixed frequencies.

This paper is organized in the following way: We introduce the lidar and the site in Sec. 2. Afterwards, interference detection
and suppression is described as well as its effects on the lidar signal evaluation in Sec. 3. Further measurements in order to
identify EMI sources are presented in Sec. 4.

## 30  2   Instruments, methods and data

The location of our "Koldeway Aerosol Raman Lidar" (KARL) is Ny-Ålesund, an international research site on Spitsbergen in
the European Arctic at $78.9°$ North and $11.9°$ East. As the Norwegian Mapping Authority Kartverket runs radio telescopes for
satellite tracking and geodetic research, Ny-Ålesund is a radio-silent village for frequencies in range $2-32\,\mathrm{GHz}$. Consequently,
using Bluetooth, WiFi devices, and radar units for airplane detection is prohibited. However, two-way radios and radio sondes
in $\mathrm{MHz}$ frequency range are used frequently and may cause RF interference.

KARL consists of a $70\,\mathrm{cm}$ mirror and a field of view of approx. $2\,\mathrm{mrad}$, a 290/50 Quanta-Ray laser from Newport-Spectra
with slightly over $200\,\mathrm{mJ}$ per pulse and color at a repetition rate of $50\,\mathrm{Hz}$. It transmits three colors simultaneously at wave-
lengths of $355\,\mathrm{nm}$, $532\,\mathrm{nm}$, and $1064\,\mathrm{nm}$. For signal detection, Hamamatsu photomultiplier (PMT), type H5573 5783-01
(www.hamamatsu.com), are used together with a gating from Licel (see details at www.licel.com). The transient recorders are
40 also from Licel (TR 20) and run both in analog (AN) and photo-counting (PCNT) mode sampling the signal with a sampling

rate of $20\,\mathrm{MHz}$. Additional lidar components are outside the scope of this paper, but a general description of KARL has been presented by Hoffmann (2011).

The weak inelastically Raman-shifted signals at $387\,\mathrm{nm}$ and $607\,\mathrm{nm}$ are sampled with $16\,\mathrm{bit}$ resolution, while the generally stronger elastic channels at $355\,\mathrm{nm}$, $532\,\mathrm{nm}$ and $1064\,\mathrm{nm}$ are sampled with $12\,\mathrm{bit}$ using old (approx. $20\,\mathrm{yrs}$) transient recorders.
In this work, we only dealt with the $532\,\mathrm{nm}$ channel (in parallel polarization) and the Lidar profile evaluation has been done according to Klett (1985).

We evaluate AN signals using PCNT signals in two steps:

1. PCNT signals have been evaluated with a lidar ratio of $\mathrm{LR} = 42\,\mathrm{sr}$ and a boundary condition of

$$\langle \beta(z_{\mathrm{ref}}) \rangle = 1.1 \cdot \beta^{\mathrm{Ray}}(z_{\mathrm{ref}}) \tag{1}$$

for altitude in the interval $24\,\mathrm{km} < z_{\mathrm{ref}} < 27\,\mathrm{km}$ to reduce the impact of an inappropriately chosen boundary condition in the lidar signals. Backscatter $\beta$ and $\beta^{\mathrm{Ray}}$ are the total and molecular (volumetric) backscatter coefficient $[\mathrm{m}^{-1}\mathrm{sr}^{-1}]$,

2. The backscatter value retrieved from this PCNT channel is then used as a boundary condition for the AN signals, where the calibration factor changes to $1.19$ compared to Eq. (1) as average in $10.5 - 11.5\,\mathrm{km}$. This boundary condition was applied to analyze the AN channel.

The lidar observations have been performed on 16 Feb 2023 between around UT 22 and UT 23 and the data is evaluated with a height resolution of approx. $7.5\,\mathrm{m}$ and an update interval of approx. $90\,\mathrm{sec}$, as the profiles of $4094$ laser shots are combined. Fig. 1 provides an overview of the lidar observations in terms of the dimensionless backscatter ratio

$$\mathrm{BSR}(z) = \frac{\beta(z)}{\beta^{\mathrm{Ray}}(z)} \tag{2}$$

of the $532\,\mathrm{nm}$ AN channel, which shows the enhancement of backscatter compared to a pure molecular atmosphere. The figure
shows clear sky conditions most of the times, a weak aerosol layer around $1\,\mathrm{km}$ altitude and finally a low cloud, which caused the end of the measurement after UT 23.

In order to identify RF interference in our setup that might disturb the lidar profiles, a Rohde&Schwarz Spectrum Rider FPH is used together with an Aaronia HyperLOG directional antenna $https://www.rohde-schwarz.com/de/produkte/messtechnik/han$ $spectrum-rider-fph-handheld-spektrumanalysator_63493-147712.html$ . Although the antenna is specified for
higher frequency band, the device delivers comprehensible (reliable and reproducible) results for frequencies around $100\,\mathrm{MHz}$.

## 3 Results

In this section, we analyze the lidar profiles in more detail with respect to disturbances and electromagnetic interference in order to evaluate the effects on the evaluation of aerosol properties. This was done since we constantly noticed an apparent and phase-constant distortion in the lidar profiles of the $532\,\mathrm{nm}$ AN channel. This distortion was omnipresent in this channel
independent of number of laser shots written in each data file. For the following discussion, only one profile from UT 22:40 is selected as example.

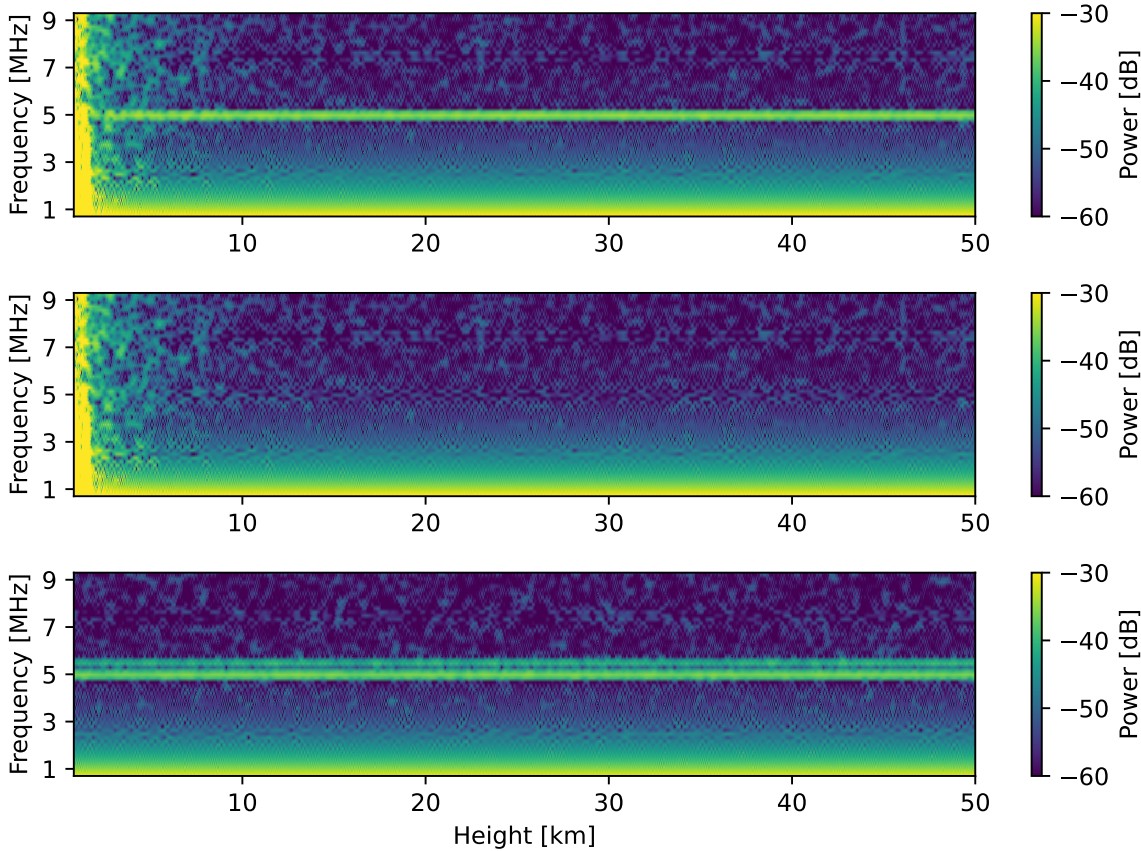

**Figure 2.** Spectrograms of the $532\,nm$ AN channel: i) before and ii) after applying the interference suppression and iii) with additional RF interference caused by using two-way radios at a frequency of $154.5\,MHz$ (recorded at another time step)

## 3.1 Detection and Suppression of RF interference

In order to detect RF interference disturbing the lidar profile, spectrograms are computed and depicted in Fig. 2 for the $532\,nm$ AN signal. For the duration of the recording of the whole lidar profile, an interference with a fixed frequency of $5\,MHz$ can be observed in the upper sub-figure. As this disturbance occurred also when all devices but the transient recorders were switched off and the coaxial cables were disconnected, we assume that this RF interference may be caused by the transient recorders themselves. The interference is present especially in the older transient recorders, so it may occur due to ageing effects of the transient recorders. In the PCNT signals, no corresponding RF interference can be observed.

Note that the signal is sampled at a sampling rate of $20\,MHz$ and the interference might occur actually at another frequency than $5\,MHz$ due to aliasing effects, which indicates that the anti-aliasing filters of the older transient recorders are somehow ineffective. While it is desirable to eliminate the interference source for future KARL measurements, e.g. by using newer

transient recorders, it is also interesting to investigate how the signal quality of existing lidar profiles might be increased by suppressing these kinds of disturbances. This is especially important due to the fact that aerosol observations with KARL have to be comparable over a long time.

In order to evaluate whether it is worth investigating in methods on how to mitigate RF interference, an easy approach for suppressing the power of single frequencies is presented. As a first step, a Discrete Fourier Transform (DFT) is applied on the lidar profile in order to determine the occurring frequency components. This is illustrated in Fig. 3 in the upper two sub-figures. A relatively weak peak can be found at a frequency of $5\,\mathrm{MHz}$, which is eliminated by cutting out eleven samples around the RF interference frequency and linear interpolating the profile at the frequencies cut out. This procedure is illustrated in Fig. 3 in the lower two sub-figures.

When applying an inverse DFT to the lidar profile after interference suppression, the spectrogram can be computed again and we expect the RF distortion to be much less visible and this is exactly what we observe in the middle sub-figure og Fig. 2. Consequently, the RF interference at $5\,\mathrm{MHz}$ is not visible anymore or at least significantly suppressed and we show in the following, how the uncertainty of the evaluation is improved by suppressing this RF interference.

## 3.2 Lidar profiles and evaluation

The lidar profiles and the corresponding evaluations are depicted in Fig. 4. Besides the (original) AN signal and the interference suppressed AN signal, the PCNT signal is also shown for comparison. In the upper two sub-figures, it can be observed that the presented interference suppression method improves the signal quality, hence the SNR, significantly. The SNR in heights between $10.5 - 11.5\,\mathrm{km}$ increases by more than a factor of 2, i.e. $3\,\mathrm{dB}$. However, the SNR improvement becomes weaker for small altitudes.

For further evaluation, we define the uncertainty $\Delta$ to determine an aerosol backscatter coefficient as

$$\Delta = \langle |\Delta\beta^{\mathrm{Aer}}(z_i)| \rangle, \tag{3}$$

where $\Delta\beta^{\mathrm{Aer}}(z_i)$ denotes the difference of the aerosol backscatter for consecutive height steps in the interval in which the AN signal was compared to the PCNT signal. The triangle brackets indicate the mean. This uncertainty $\Delta$ decreased from $1.56 \cdot 10^{-7}$ to $7.1 \cdot 10^{-8}$ (units: $m^{-1}sr^{-1}$) by the presented RF interference suppression . Although the PCNT signal has still a higher SNR and with $\Delta = 3.1 \cdot 10^{-8}$ also a lower uncertainty, the AN signal improvements are useful when both channels are combined. As in our case the RF interference manifests in a periodic disturbance of the lidar signal which can be background corrected, it does not introduce a bias in the retrieval of aerosol properties. However, as (Veselovskii et al., 2002) and Boeckmann2001 have pointed out, an inversion of microphysical aerosol properties from multiwavelength lidar requires an uncertainty of the optical coefficients of less than 10 %.

In the lower two sub-figures of Fig. 4, the backscatter ratio BSR and the aerosol backscatter $\beta^{Aer}$ are illustrated. The improvement by the presented RF interference suppression can be observed especially for heights above $7.5\,\mathrm{km}$. The low values of the PCNT signal below $2\,\mathrm{km}$ altitude are due to signal saturation.

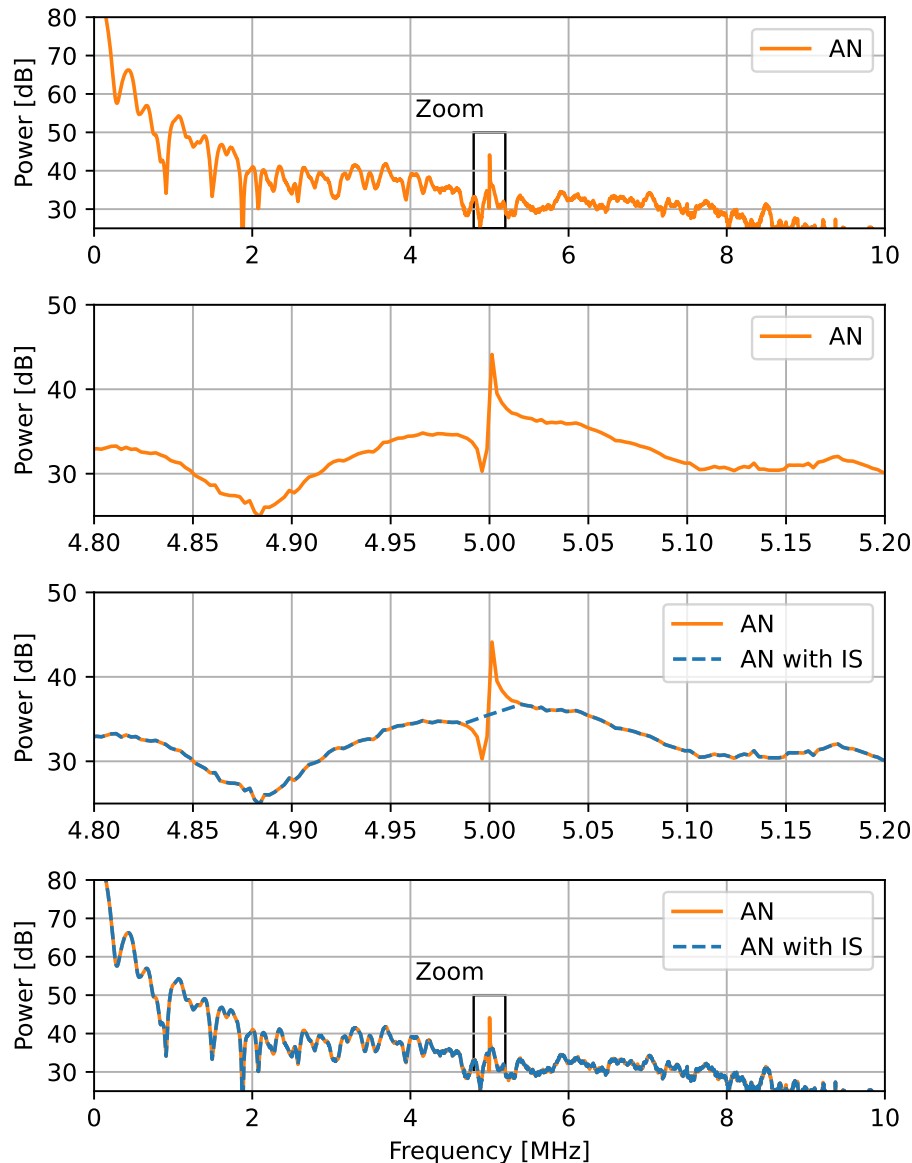

**Figure 3.** Illustration of the interference suppression (IS) applied to the $532\,\text{nm}$ AN channel: i) Spectrum with weak interference peak at $5\,\text{MHz}$, ii) weak interference peak, iii) with linear interpolation between two samples as an IS method, and iv) the spectrum with IS

## 4    Identification of other interference sources

When analyzing the lidar profiles, we noticed also other RF interference sources in our setup. Since two-way radios are used in Ny-Ålesund for communication purposes, we evaluated if these might be the reason for distortions in the lidar profiles

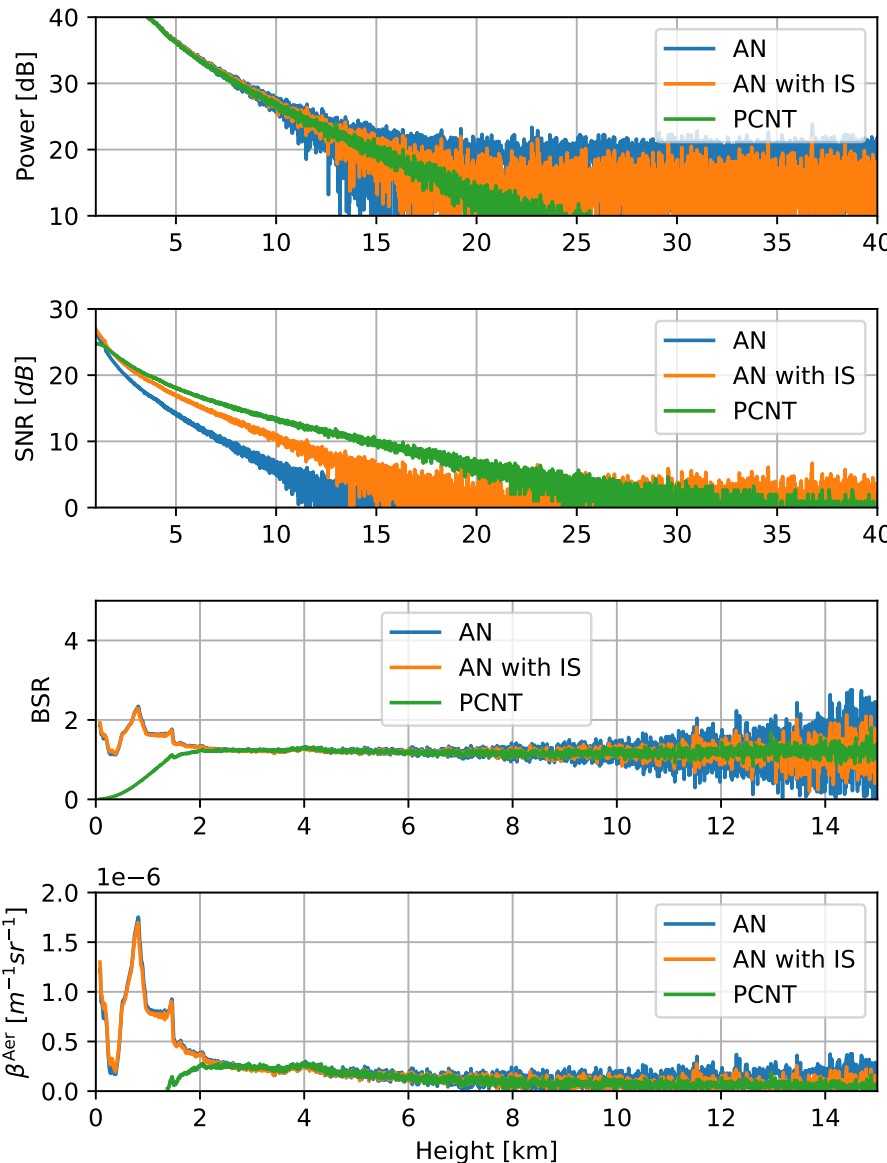

**Figure 4.** Evaluation of the $532\,\text{nm}$ channel with the AN signal with and without IS as well as the PCNT signal: i) Lidar profiles, ii) Signal-to-noise ratio SNR, iii) Backscatter ratio BSR, and iv) aerosol backscatter coefficient $\beta^{\text{Aer}}$

occurring from time to time. Thus, we radioed continuously during a measurement and the Rohde&Schwarz Spectrum Rider FPH confirmed that the radio channel was located at $154.5\,\text{MHz}$. The corresponding spectrogram of the lidar profile is depicted in the lowest sub-figure of Fig. 2. It can be observed that there is an additional frequency at $5.5\,\text{MHz}$ disturbing the lidar profile. This perfectly fits to the two-way radio channel as this will occur at $5.5\,\text{MHz}$ when sampling with $20\,\text{MHz}$ due to aliasing.

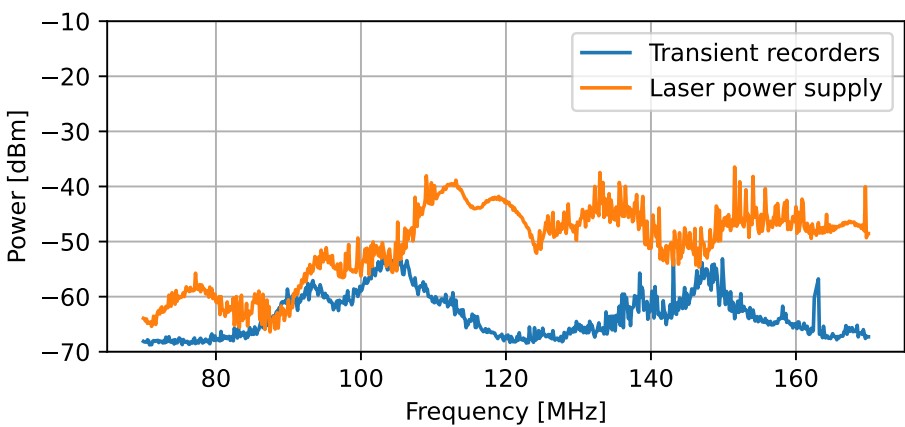

**Figure 5.** Spectrum of EMI in the room housing the transient recorders and the room with the laser and its power supply.

Further, we measured the electromagnetic environment of the laser power supply and the transient recorders in order to evaluate if there might be also electromagnetic interference caused by the power supply. The measurements are illustrated in Fig. 5 and it can be seen that the RF power in the observation room is approx. $20\,\mathrm{dB}$ lower than inside the room with the laser power supply. For these measurements the Spectrum Rider has been placed approx. $50\,\mathrm{cm}$ away from either the rack containing the transient recorders and the laser power unit respectively. However, the precise distance turned out to be uncritical because the radiation did not show apparent gradients in each room. Nevertheless, if the transient recorders would be placed directly next to the laser power supply, there might occur electromagnetic interference due to laser operation. Fortunately, our setup in Ny-Ålesund has a separated observation room and a clear impact of the laser unit on the transient recorders has not been found.

## 5    Conclusions

Although we would expect no RF interference to happen in a radio-silent area, it occurs in the lower frequency range especially for geometrically large lidar systems with long cables. In this paper, we presented how RF interference can be detected by applying spectral analysis to the lidar signals and we also provided an easy method for interference suppression. We found out that even suppressing the relatively weak interference has positive effects on the lidar evaluation. Especially weak signals from higher altitudes in ground-based systems benefit from interference suppression. The frequency-selective interference occurred in the analog channel of the old transient recorders and could be suppressed using the presented method.

Finally we presented measurements indicating that placing the power supply of the laser next to the transient recorders may also lead to electromagnetic disturbances in the lidar profiles. In our case, the placement of the laser in a room separated from the transient recorders reduces the disturbances significantly. However, electromagnetic compatibility has to be taken into consideration in order to obtain high quality data. Consequently, further work has to be done in order to make KARL more robust against external influences.

For a trustful retrieval of microphysical properties of aerosol from lidar data, backscatter and extinction coeficients must be recorded with less than 10 % uncertainty (Veselovskii et al., 2002), (Böckmann, 2001). Hence, at least sporadic checks on the RF interference occurrence in analog signals is recommended. In case interference occurs with fixed phase shift, a dark signal correction is preferred over a simple background correction. However, if the RF interference results from external sources

appearing at fixed frequencies, it should be filtered out as described.

*Code and data availability.* The entire used data set and software used for evaluation is available on request.

*Author contributions.* This work was jointly performed by both authors who also wrote the manuscript together.

*Competing interests.* The authors declare no conflict of interest.

*Acknowledgements.* The lidar was serviced at AWIPEV station, Ny-Ålesund, by Wilfried Ruhe, who also supported the measurements.
Special thanks to Rubén Bolaño Gonzáles from Kartverket for borrowing the device for spectral measurements.

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
