# Peer review of "Influence of Electromagnetic Interference on the evaluation of Lidar-derived Aerosol properties from Ny-Ålesund, Svalbard"

_Atmospheric Measurement Techniques, 2023_

## Author Response (AR1)

Changes in manuscript:

All changes in the manuscript (compared to previous version for reviewers) are marked in red.

The new blocks are:

End of introduction:

However, de-noising techniques in lidar from a general point of view are outside the scope of this paper. Due to the strong dependence of SNR on altitude such a noise filtering is commonly done by wavelet filtering (Zhou et al., 2013) or (Mao, 2012). Instead, we will show that EMI can be suppressed in frequency domain, if it appears at fixed frequencies.

Instruments, Data and Methods (after eq 1)

for altitude in the interval 24 km < zref < 27 km to reduce the impact of an inappropriately chosen boundary condition in the lidar signals. Backscatter $\beta$ and $\beta$Ray are the total and molecular (volumetric) backscatter coefficient [m−1sr−1]

last paragraph:

In order to identify RF interference in our setup that might disturb the lidar profiles, a Rohde&Schwarz Spectrum Rider FPH is used together with an Aaronia HyperLOG directional antenna https : //www.rohde−schwarz.com/de/produkte/messtechnik/han

spectrum − rider − f ph − handheld − spektrumanalysator63493 − 147712.html . Although the antenna is specified for higher frequency band, the device delivers comprehensible (reliable and reproducible) results for frequencies around 100 MHz.

Results:

First paragraph:

In this section, we analyze the lidar profiles in more detail with respect to disturbances and electromagnetic interference in order to evaluate the effects on the evaluation of aerosol properties. This was done since we constantly noticed an apparent and phase-constant distortion in the lidar profiles of the 532 nm AN channel. This distortion was omnipresent in this channel independent of number of laser shots written in each data file.

3.1: second paragraph:

Note that the signal is sampled at a sampling rate of 20 MHz and the interference might occur actually at another frequency than 5 MHz due to aliasing effects, which indicates that the anti-aliasing filters of the older transient recorders are somehow ineffective.

3.2, beginning:

The lidar profiles and the corresponding evaluations are depicted in Fig. 4. Besides the (original) AN signal and the interference suppressed AN signal, the PCNT signal is also shown for comparison. In the upper two sub-figures, it can be observed that the presented interference suppression method improves the signal quality, and hence the SNR, significantly.

3.2, behind eq.3:

where $\Delta\beta_{Aer}(z_i)$ denotes the difference of the aerosol backscatter for consecutive height steps in the interval in which the AN signal was compared to the PCNT signal. The triangle brackets indicate the mean. This uncertainty $\Delta$ decreased from $1.56\cdot10^{-7}$ to $7.1\cdot10^{-8}$ (units: $m^{-1}sr^{-1}$) by the presented RF interference suppression. Although the PCNT signal has still a higher SNR and with $\Delta = 3.1 \cdot 10^{-8}$ also a lower uncertainty, the AN signal improvements are useful when both channels are combined.

As in our case the RF interference manifests in a periodic disturbance of the lidar signal which can be background corrected, it does not introduce a bias in the retrieval of aerosol properties. However, as (Veselovskii et al., 2002) and Boeckmann2001 have pointed out, an inversion of microphysical aerosol properties from multiwavelength lidar requires an uncertainty of the optical coefficients of less than 10 %

Conclusions, end:

For a trustful retrieval of microphysical properties of aerosol from lidar data, backscatter and extinction coefficients must be recorded with less than 10 % uncertainty (Veselovskii et al., 2002), (Böckmann, 2001). Hence, at least sporadic checks on the RF interference occurrence in analog signals is recommended. In case interference occurs with fixed phase shift, a dark signal correction is preferred over a simple background correction. However, if the RF interference results from external sources appearing at fixed frequencies, it should be filtered out as described.

Ref 3

We thank the reviewer for his comments. Please find our answers in red (here and in the manuscript)

While the manuscript provides valuable insights into the influence of electromagnetic interference on lidar-derived aerosol properties, there are a few weak points that could be addressed to improve it:

Limited discussion on the potential impact of RF interference on aerosol property retrieval: The manuscript focuses primarily on the detection and suppression of RF interference but provides limited discussion on how such interference affects the accuracy and precision of aerosol property retrieval. Further exploration of the potential biases or uncertainties introduced by RF interference would enhance the manuscript.

We don't expect a bias, as the RF interference manifests as a (false) oscillatory behaviour of the lidar signal. We will state this in the new version. However, this RF interference increases the uncertainty, as we explained in section 3.2: the uncertainty of beta_aer decreases by a factor of (slightly more than) 2 in our case at 10.5 to 11.5km.   For a retrieval of aerosol properties typically uncertainties of beta_aer below or equal 10% are required. The according quotes of Veselovskii et al. and Böckmann are given in the new version of the manuscript. Hence the improvement by RF interference detection is most obvious for weak aerosol layers.

See also our answer to Reviewer # 5. We provide a plot on the uncertainty over altitude.

[Figure]

However, as the uncertainty depends on the meteorologic conditions (the signal strength) no general conclusions can be drawn from that plot.

Böckmann, C. 2001, Appl. Opt. https://opg.optica.org/ao/abstract.cfm?uri=AO-40-9-1329

Veselovskii I. et al. 2002, Appl. Opt.  https://opg.optica.org/ao/abstract.cfm?uri=ao-41-18-36852002,

Lack of comparison with other lidar systems: The study primarily focuses on the lidar system used in Ny-Ålesund, Svalbard, but does not provide comparisons or discussions about similar lidar systems in other locations or studies. Including such comparisons would strengthen the significance and generalizability of the findings. Maybe this point can just be indicated as future research.

Thank you for this remark. Indeed, our findings can be generalized to any analog signal. Of course, we are open to any cooperation in the field. In the past even a few intercomparison campaigns have been performed in Ny-Ålesund, even if it is logistically demanding.

Limited exploration of noise reduction techniques: While the manuscript presents an interference suppression method, it does not explore other noise reduction techniques commonly used in lidar data analysis, such as wavelet filtering or advanced denoising algorithms. Discussing the limitations and potential improvements of the proposed interference suppression method in comparison to existing techniques would add depth to the manuscript.

This is a valid point. The importance (and accuracy) by noise reduction is probably a very important topic. However, we would avoid a long discussion here for two reasons: First, we believe that this issue is best done by artificial lidar signals (where the exact solution is known but hidden behind some noise). In this manner the different smoothing / de-noising techniques could be compared best. Second, if someone sees a clear artificial spike in the Fourier space, it is probably best to correct this spike directly. This is what we want to show by our manuscript: regardless of how the lidar evaluation is done, if RF interferences are detected it is worth to correct them directly.

Insufficient discussion on the implications for long-term data records: The manuscript briefly mentions the importance of long-term data recording and quality assurance. However, further discussion on the implications of RF interference on long-term data records, including the potential biases or uncertainties introduced over time, would provide valuable insights for researchers relying on lidar data for climatological studies.

Thanks. We will mention the quotes Böckmann 2001 and Veselovskii et al. 2002 who state that the optical coefficients are required with less than 10% uncertainty for trustful inversion of microphysical aerosol properties. See also below (answer to Rev 5.) We add an additional plot on how the uncertainty decreases for our case on 16 Feb. However, the uncertainty reduction depends on the strength of the lidar signal and hence on the meteorologic conditions. Therefore, there is no easy answer and this additional plot is only an example which may not be included in the final manuscript.

Rev # 4

We thank the reviewer for his comments. Please find our answers in red (here and in the manuscript).

The paper is very interesting because it deals with electromagnetic interferences that is often a problem that affects lidar systems, and the related discussion is still going on within the lidar community. Therefore, further studies on this topic are always welcome.

The paper is well written. Just some comments:

1) concerning fig.4, it would be important show also the ratio between the corrected and not corrected profiles both for the aerosol Power (i) and for the aerosol backscatter (iv), because it is not possible to evaluate the differences and draw conclusions just looking at the profiles.

We show the ratio between the corrected and uncorrected aerosol backscatter profiles for a new case with cloud, please see below.

2) did the authors check the influence of the filter on the backscatter profile also in in presence of a strong aerosol layer? This would be important to evaluate the effect of the filter in presence of sharp and strong changes in the measured signals.

We have checked it as required. As expected, the method works, see the plots below. This is because a sharp edge of cloud bottom / top consists of many different frequencies in Fourier space. Hence

filtering out one corrupt frequency does not change the solution. You can see in the figure below that the cloud is basically unaffected by filtering.

[Figure]

The next figure shows, as requested, the ratio between the original (raw) and filtered aerosol backscatter profile.

[Figure]

As 5MHz in a lidar refers to 30m you can see the oscillation of 30m in the plot below whenever the backscatter is very low. When the backscatter becomes larger, as in the cloud, the two solutions are basically identical, hence the ratio of the aerosol backscatter (raw / filtered) is close to 1. If requested we can show these plots in an attachment.  (However we think that the results are as expected.)

The equivalent plot as the equivalent to Fig 4 in the manuscript look like this

Spectrum without cloud:

[Figure]

And the spectrum with cloud:

[Figure]

3) did the authors try to measure the dark signal and subtract it to the measured signal to compare the results?

We thank the reviewer for this comment! This is useful for the community. We only subtract background counting rates from signals > 60km altitude. As the RF interference originates from the transient recorders it occurs at the same height intervals in our case. Hence by a dedicated dark signal subtraction the impact of the RF interference can be reduced. (If it is strictly constant over time): We will clarify this in the new version of the manuscript. However, this assumes that environmental EM sources are strictly constant. Other users whose RF sources may came from external sources like two-way radios need to suppress the RF interference like stated in the manuscript. We will point this out in the new version.

Rev # 5

We thank the reviewer for his comments. Please find our answers in red (here and in the manuscript).

Review of  AMT manuscript amt-2023-79

Title: Influence of Electromagnetic Interference on the evaluation of Lidar-derived Aerosol properties from Ny-Ålesund, Svalbard

Author(s): Tim Poguntke and Christoph Ritter

The messages of the manuscript are:

1. A 5 MHz interference has been detected in the analogue signal of the lidar system KARL.

2. It is suspected that the interference comes from the data acquisition itself, due to ageing effects.

3. The KARL measurements could be impaired since a long time, and the correction of the old measurements would be beneficial.

I guess that future publications using the old KARL measurements need a reference to how the distortion has been corrected.

Yes, thanks. We will filter the noise frequency exactly like in section 3 (Fig 3) of this work. While in principle any proper noise filtering will yield "similar results" by judging the resulting aerosol backscatter profile, it seems natural to subtract the impact of one (or a few) corrupted individual frequencies once they are identified in the data.

4. A simple method for the correction of old measurements by means of digital frequency filtering is presented.

5. Another interference signal at 5.5 MHz has been identified when two-way radios sending at 154.5MHz were used.

6. A wide EMI spectrum has been measured close to the laser power supply.

For other lidar users it would be interesting to read about the following:

- How strong is the 5.5 MHz signal in other than 4094 laser shot averaged signals?

As we wrote in the manuscript, for our system with the negligible external EM sources the RF interference seem to be in phase with respect to the laser. As you can see from the plot below, the noise power is the same regardless of laser shots per file. However, as this is a specific result of our system (which can be different in other lidars) we would not necessarily show the plot in the manuscript. The reason is that in more then 30km altitude in polar night in the analog signal non-optical noise dominates, which, in our case is mainly the 5MHz frequency. For this reason we see this odd 30m oscillation.

[Figure]

If the strength relative to the lidar signal does not change with averaging, the interference signal must be in phase with the laser repetition.

Yes, this is the case for our system, see above

If it is much stronger in the single shot signal, which indicates random phase with respect to the laser, it could cause more distortions than just a SNR reduction.

This is a good point, even if this seems not to be the case in our system we will mention this in the new version of the manuscript.

- How big is the influence of this interference on the lidar signal products like backscatter and extinction coefficient?

We now quote Veselovskii et al. 2002 and Böckmann 2001 who say that for a retrieval of microphysical properties of aerosol the optical coefficients need to be known with max. 10% uncertainty. An example for RF interference induced noise is given in section 3.2. The precise amount of induced uncertainty depends on the strength of the aerosol layer. As the lidar signal is stronger close to the ground and in thicker aerosol layers the uncertainty under these conditions decreases. This means that no "one fits all" uncertainty calculation can be given. For our case on 16 Feb 2023 the uncertainty introduced by RF interference looks like this. We prefer to not show this figure in the final paper because, as stated above, this curve depends on the environmental conditions.

[Figure]

An analysis of more than just one example would show the importance of the correction.

Please see our answer to the reviewer # 4. (Our method works in cases of strong layers like clouds because strong layers have well defined bottom and top altitudes. Hence, a layer consists of many frequencies in Fourier space. Filtering out one or few corrupt frequencies will not change the solution).

- Which parts of the data acquisition are suspected of ageing and as reason for the decreased suppression of the anti-aliasing filter?

We cannot say more than we know that anti-aliasing filters are used even in the oldest transient recorders from licel and that they usually work as we have not seen this behaviour when the transients were younger.

Has the manufacturer been consulted?

Only briefly – although we generally have a good relation to Licel. See below

Can it be repaired?

We had this and other transient recorders for a checkup at Licel about 10 yrs ago. This was much cheaper than to buy new transient recorders. However, now after 20 yrs of operation we are seriously considering an upgrade to completely new transients.
* * *
Detailed questions and proposals:

We thank the reviewer for all the following corrections! Your suggestions are included in the new version.

Line 26: Proposal: " ... we will show that, if EMI is present and if it manifests itself at fixed frequencies, it can be suppressed in the frequency domain."

Line 36: Proposal: KARL consists of a telescope with a mirror of 70cm diameter and a field of view...

Line 36: I couldn't find in the internet a "290/50 Spectra laser". Please detail that.

https://www.laserlabsource.com/files/pdfs/solidstatelasersource_com/product-305/Nd_Yag_Laser_Nanosecond_Laser_1064nm_1250mJ_Spectra_Physics-1462086952.pdf

Line 38: "Hamamatsu photomultiplier (PMT)":  because the photomultipliers can also be an antenna for EMI, it would be interesting to know more details about the model.

It is an H5573 5783-01 . Note that we see the disturbance only in the AD signal not in the counting.

Please provide a reference for "Hamamatsu".

Line 38: "..with a gating from Licel."  What does this mean? Please explain. Please provide a reference for "Licel".

Please see   https://licel.com/manuals/pmtmanualgating.pdf

Line 42:  Proposal: " ... are sampled with 16bit resolution,..."

Eq. (1) and line 49:  How is the boundary condition of backscatter ratio 1.1 motivated and why does it "reduce the impact of noise in the lidar signals" ?

1.1 is our "clear sky approximation" which is a generally justified value for our site, see also https://www.mdpi.com/2072-4292/14/11/2578

And sorry, noise is the wrong word!! We mean that ir reduces the uncertainty of beta_aer (as pointed out by Klett – "backward integration)

Line 50: Proposal: " β and β_Ray are the total and molecular (volumetric) backscatter coefficients [ m– 1 sr– 1 ]."

Line 51f: Proposal: " The backscatter ratio retrieved from the PCNT channel at 10.5 - 11.5 km is then used as a boundary condition for the AN signal there."

Line 59:  "...a weak aerosol layer below 1km altitude...": the height below 1 km is not shown in the plot.

We changed it to "around 1km" – the lower "edge " if Fig 1

Line 60ff:  Please provide references for  "Rohde&Schwarz Spectrum Rider FPH"  and " Aaronia HyperLOG directional antenna".

Line 63:  "... device delivers comprehensible results ...": Please specify what you mean with "comprehensible results".

We mean reliable and reproducible

Line 67ff: "... one profile from UT 22:40..." and

Line 71:   "...the duration of the whole lidar profile...": please specify (again) the duration of the lidar profile.

Thanks! We changed duration of the recording of the whole lidar profile (120km in total, no pretrigger)

Line 72:  How are the transient recorders triggered if everything is switched off? Is this triggering different from the normal measurement situation?

We have an external trigger generator fr test like this

Line 74f: Why do you suspect "ageing effects" as a source for the interference? Which part is considered as possibly ageing?

As stated in the manuscript: we have not seen this disturbance in older data and the other (younger) transient recorders do not show this behaviour.

Line 76ff: "... which indicates that the anti-aliasing filters of the older transient recorders are somehow ineffective....": This deduction is not coercive. Please explain.

We saw in the licel.com internet pages that all their transient recorders are equipped with anti aliasing filters. Another reviewer wanted to have a speculation about this. It is likely that here a problem occurred, even if we cannot prove this. Therefore "indicate". Our point is generally that it is worth to check the raw data quality every now and then.

Fig. 3 caption: i), ii) ... is not unambiguous.

This remark is not clear to us, unchanged

The interpolation seems to be over more than two samples. Actually, in the text is mentioned that you "cut out" eleven "samples". In the frequency domain the spacing of frequency points depend on the resolution of the DFT. Is a filter applied? Please explain in more detail.

As explained in the Fig caption we linearly interpolated between all the last trusted frequency before and the first behind the disturbance. Simply as shown in Fig 3 iii

Line 91: "... how the evaluation is improved ...": Which evaluation?

We changed to the uncertainty of the evaluation

Line 95: Proposal: "... the presented interference suppression method improves the SNR significantly."

Line 97ff: Proposal: exchange insecurity by uncertainty.

Line 99ff: "... the difference of the aerosol backscatter from one height to the other in the interval in which the AN signal was compared to the PCNT signal. The triangle brackets indicate the mean. " What does that mean? Equation (3) and the explanations are not understandable.

Thanks. We changed the wording.

Line 103f: "The improvement by the presented RF interference suppression can be observed especially for heights above 7.5km." What improvement do you observe?

See the provided plot above

Line 112: " … occur at 4.5MHz … " Why at 4.5 MHz?

Thank you. That was a typo. We mean 5.5MHz (8*20-154.5)

Line 113: What is the "electromagnetic environment of the laser power supply and the transient recorders" ?

The power of the existing EM radiation as presented in Fig 5.

Line 122f: "… it occurs in the lower frequency range especially for geometrically large lidar systems with long cables."

But in Line 72f you write: " As this disturbance occurred also when all devices but the transient recorders were switched off and the

coaxial cables were disconnected, …" So the cables are not the antennas that pick up the interference. Right?

Right. In our systems we can now rule out that cables act as antennas. For other systems this may be a concern though.

Line 125f: " Especially weak signals from higher altitudes in ground-based systems benefit from interference suppression." How do they benefit?

We refer to Fig 4 (blue vs orange signals)

Line 128f: "Finally we presented measurements indicating that placing the power supply of the laser next to the transient recorders

may also lead to electromagnetic disturbances in the lidar profiles."

Your measurements don't indicate that. While it is always possible, that RF noise can disturb measurements, it is not clear whether the RF power you measured close to the laser power supply would really disturb the signals of the transient recorders. An experimental proof or reference is missing.

We think that our wording is justified. Look at Fig. 5: next to the laser we have much more EM noise especially in the 110Mhz to 160MHz range, in which also the 2-way radios are operating for which the lidar electronics are susceptible.